# Dealing with the Ambiguity of Glycan Substructure Search

**DOI:** 10.3390/molecules27010065

**Published:** 2021-12-23

**Authors:** Vincenzo Daponte, Catherine Hayes, Julien Mariethoz, Frederique Lisacek

**Affiliations:** 1Department of Computer Science, University of Geneva, 1227 Geneva, Switzerland; catherine.hayes@unige.ch (C.H.); julien.mariethoz@sib.swiss (J.M.); 2Proteome Informatics Group, SIB Swiss Institute of Bioinformatics, 1211 Geneva, Switzerland; 3Section of Biology, University of Geneva, 1211 Geneva, Switzerland

**Keywords:** glycan structure, knowledge representation, pattern recognition, ontology, semantic web

## Abstract

The level of ambiguity in describing glycan structure has significantly increased with the upsurge of large-scale glycomics and glycoproteomics experiments. Consequently, an ontology-based model appears as an appropriate solution for navigating these data. However, navigation is not sufficient and the model should also enable advanced search and comparison. A new ontology with a tree logical structure is introduced to represent glycan structures irrespective of the precision of molecular details. The model heavily relies on the GlycoCT encoding of glycan structures. Its implementation in the GlySTreeM knowledge base was validated with GlyConnect data and benchmarked with the Glycowork library. GlySTreeM is shown to be fast, consistent, reliable and more flexible than existing solutions for matching parts of or whole glycan structures. The model is also well suited for painless future expansion.

## 1. Introduction

Glycosylation is an important modification of proteins and lipids. It entails the attachment of a broad variety of glycan molecules to form glycoconjugates. A large variety of experimental techniques can be used to solve glycan structures [1], and the higher the throughput, the lower the resolution. Consequently, glycan data accumulate heterogeneously with uneven levels of precision. A glycan structure repository called GlyTouCan [2] collects this information online and assigns a stable accession number to each glycan irrespective of its degree of characterization.

Glycans are branched tree-like molecules composed of a few hundred monosaccharides according to MonosaccharideDB (Available online: http://www.monosaccharidedb.org (accessed on 1 November 2021)). Roughly a dozen of these cyclic building blocks are very frequent in known structures, such as mannose, glucose, and galactose. These monosaccharides are linked together in varying ways, depending on carbon attachment positions in the cycle, to form a full structure. A glycan remains, often described in the IUPAC linear format [3], delineating branching structures with different bracket types. This encoding is known to generate directional/linkage/topology ambiguity and as such is not sufficient to fully represent incompletely characterized molecules that may contain repeated units. The limitations of the IUPAC linear format led to the definition of several concurrent formats exploiting the tree/graph-like nature of glycans. In a graph representation of a glycan, each monosaccharide is a node possibly associated with attributes, and each linkage is an edge also potentially associated with attributes. The chemical bonds between building blocks, designated as glycosidic linkages, define the edges of the acyclic graph structure.

Several graph-based encoding schemes have been proposed, e.g., Glyde [4], IUPAC condensed [5], KCAM [6], KCF [7], GlycoCT [8], or more recently WURCS [9]. This uncoordinated production and usage gave rise to a resourceful converter tool called GlycanFormatConverter [10] that facilitates data submission to GlyTouCan. In GlyTouCan, an entry can be a simple composition (a list of monosaccharides) or can include topologies in which no carbon position is detailed and lacking anomeric configuration. In order to cope with this ambiguity, the Glycan Naming and Subsumption Ontology (GNOme) was developed (available online: http://www.obofoundry.org/ontology/gno.html (accessed on 1 November 2021)). GNOme relies on GlyTouCan entry identifiers to guide data browsing in both GlyToucan and GlyGen [11] databases.

Programmatic access to GlyTouCan mainly depends on the GlycoCT and WURCS formats, and the former is commonly used as a data sharing format across other glyco-related databases. The display of structures complies with the Symbol Nomenclature for Glycans (SNFG) nomenclature [12,13], now recommended in all domains of glycoscience. This notation assigns each monosaccharide to a coloured shape (e.g., yellow circle for galactose, shortened as Gal). Shared colours or shapes express structural similarity among monosaccharides. For example, N-Acetylgalactosamine (yellow square) differs from galactose (yellow circle) through a so-called substituent (removal of an OH group and addition of an amino-acetyl group). “Substituent” as a property is precisely the type that qualifies a node.

Searching (sub)structures is a routine exercise in structural glycoscience. Among other things, it solves the search for glycan epitope (or determinant) that delineates the binding part of the whole structure targeted by glycan-binding proteins. At this point, we distinguish our definition of substructure search from motif finding, which was recently reviewed in [14]. Motif finding methods are primarily implemented to capture glycan-binding specificity mainly in array data and take advantage of affinity binding to select and extract glycan substructures defining a ligand. The question addressed here targets the overall relatedness between glycan structures. It does not assume the existence of motifs and provides the means of comparing, grouping, or browsing structures independent of the quality of their resolution.

In this article, we introduce an ontology modelling a tree-logical structure to represent glycans from compositions to fully resolved structures. The proposed method, called GlySTreeM, builds on the previous GlyS3 tool [15] developed in SPARQL [16] with which knowledge bases defined using the Resource Description Framework (RDF) can be queried. GlySTreeM is validated with the content of the GlyConnect database [17] and benchmarked with the Glycowork Python library [18] in which IUPAC strings are translatable into graphs that can be processed as such.

## 2. Materials and Methods

### 2.1. Approach

At present, one of the most accepted formats for the storage of glycan structures is GlycoCT, a connection-table type representation that is both human- and computer-readable. A GlycoCT string encodes structural information and in theory, assigns a unique code to each glycan. This format is used in GlyConnect, our in-house database that collects glycoproteins in association with glycan data. Currently, the RDF representation of GlyConnect relies on the GlycoCoO ontology [19]. However, this ontology does not cover individual structures themselves. An initial exploratory study identified the use of RDF syntax as the most appropriate for the representation of glycan structures [15]. While this model shows potential, it has two main flaws: (1) searching for structures could not be extended to compositions and (2) the visual model (SNFG nomenclature) was disconnected from the knowledge model (GlycoCT), thereby challenging the decomposition of one-to-many relationships [20]. The following describes an updated translation of the structure format to allow for quicker and less computationally expensive mapping and importing of structures, whether fully or partially defined.

The main goal of this system is to efficiently search precise or ambiguous structures or substructures, which involves the following tasks:A comprehensive representation of the structure of glycans;In-depth access to the data from various logical perspectives;A navigation method that allows the exploration of the structure as well as the facility to query through substructures.

The achievement of this goal requires the conception of a model that provides the necessary features of expressiveness and soundness. To this aim, the ontology was chosen as an instrument for the reification of such a model. An ontology representation of the glycan structure may provide the necessary expressiveness to represent the various aspects of these structures as well as ensure interoperability between this and external resources [21].

### 2.2. Model Design

The key idea behind the conception of the new model is to separate the glycan structure from the building blocks that compose it. This approach aims to enhance the exploration perspectives of the glycan structure. To achieve this goal, the knowledge system is designed to provide the semantic tools to (1) represent deeper aspects of both the structures and the building blocks, (2) allow a substructure navigation of the individuals represented, and (3) query for incomplete structures.

The first step in formulating a new model was the deconstruction of GlycoCT strings [22]. Previously, each discrete piece of information was parsed from the GlycoCT string and saved in the model. In GlycoCT format, HexNAc (or any of its family members, GlcNAc, GalNAc, etc.) is made up of two building blocks: a base, Hexose, and a substituent, N-acetyl, (Figure 1). These have a specific linkage that identifies the overall residue as a “combined monosaccharide”, which in turn is represented as a square in SNFG format (cartoon). The syntax notations used to represent the GlcNAc building block according to IUPAC and GlycoCT standards are shown in Table 1.

In-depth access to the different parts of the glycan requires these parts to be semantically defined together with logical bounds that link them. To this end, the logical parts that compose the glycan structure were identified as well as the logical links that bind them. The model is discussed starting from the higher abstraction level to go deeper into the details (as zooming into a map).

At the highest abstraction level lies the *Glycan* class. It represents the high-level class in the glycan abstraction, together with its general properties. All the resources referring to a structure are related either directly or indirectly to it. Linked to this class are the main Glycan components identified with the *GlycanCore* class, linked through the *hasGlycanCore* relationship, and the *GlycanBag* class linked through the *hasGlycanBag* one. These relationships are both asymmetric and functional and decorated with domain and range axioms; in particular, a *Glycan* can be linked to only to a *GlycanCore* and a *GlycanBag*, as shown in Figure 2.

The *GlycanCore* represents the unambiguously defined part of the structure, known as the *core* represented as a tree with a root and multiple children per node. The tree structure allows the representation of all the residues that compose the core of the glycan.

The *GlycanBag* class represents the set of the substructures for which the link with core is unknown. This class represents a set of loosely defined single residues but also of substructures (sub-trees). These unknown-linked structures are considered as items of the bag, and each item is related to the root of each substructure.

The *Residue* class represents the node of the tree structure, both in the core and in the bag items. It is related to the building block configurations without holding the information, hence keeping them logically decoupled from the structure. Since each tree is rooted (core and bag items), the *ResidueRoot* class was created to identify the root of the tree structure. This class is defined as a *subClassOf* the *Residue* class, as shown in Figure 2. To widen the representation of the building block structure and increase the flexibility of the search, the *Residue* class has been decorated with the *composition* data property. This optional data property describes a group of isomeric monosaccharides or a substituent residue according to the weight of the building blocks (base and/or substituent) that compose it.

To represent the tree nature of the glycan structure and to ensure an efficient navigation, we relied on a recently described ontology modeling a tree logical structure [23]. This ontology provides the logical wrappers to capture the key elements of a tree. For example, the *TreeNode* and the *RootNode* classes are used to wrap—as super-classes—respectively, the *Residue* and the *ResidueRoot* classes. To represent the group of undefined residues, the bag logical model that belongs to the same library was chosen. Therefore, the *GlycanBag* class has been declared as a subclass of *Bag*, and the *GlycanBagItem* class as a subclass of the *BagItem* class, inheriting all the axioms and properties related to their logical ancestors.

The defined structure of the glycan can then be extended with building blocks populating the glycan. The building blocks are decomposed, following the GlycoCT syntax, into bases and substituents, which are represented, respectively, through the *Base* and *Substituent* classes. The *Residue* class is linked to these classes through the *hasBase* and *hasSubstituent* relationships. The *Base* and *Substituent* classes collect the building block information on:Name of the building block.Anomeric connection.Parent building block anomeric connection.Anomer configuration.

The decision to use “composed rules” for a set of residues is based on the activated sugar donors in animal cells, which includes GlcNAc, GalNAc, NeuAc, etc. [12]. The residue representation of the GlcNAc building block is taken as an example to show how GlySTreeM represents such residues. The source of knowledge is the GlycoCT string reported in Table 1, which corresponds to the GlySTreeM representation shown in Figure 3.

### 2.3. Implementation and Data Mapping

GlySTreeM knowledge base was populated with GlyConnect structures encoded in GlycoCT. To provide a flexible data import, the pipeline was divided into four parts: (1) the source wrapper that handles access to the GlycoCT string of the structure, (2) the algorithm that translates GlycoCT strings into GlySTreeM individuals, (3) the rule system that generates the *composed* residues, and (4) the rule system that assigns a composition to the residues.

(1) The source wrapper was designed to receive and parse data from the JSON API of GlyConnect and to directly access the GlycoCT strings. Such a module can be replicated and adjusted to receive data from any other source collecting structures in GlycoCT, without affecting the other parts of the pipeline.

(2) The algorithm that parses the GlycoCT strings begins from the decomposition of the structure into defined mandatory or optional sections. The mandatory describe the residues and the linkages, while the optional describe undefined portions of the structure.

(3) After creating the high-level classes (Glycan, GlycanCore and GlycanBag) individuals, the algorithm explores the linkages listed in the GlycoCT string to build the residue structure and decorate them with the bases and substituents declared in the GlycoCT *RES* list. At this stage, the rule system for composed residues provides the indication of whether to create one residue hosting two building blocks (composed) or one residue per building block. The rules are defined as JSON objects, and for each object, the following is defined: (a) the first building block (usually a base) name, (b) the second building block (usually a substituent) name, (c) the carbon number of the first building block involved in the linkage, (d) the carbon number of the second building block involved in linkage, and (e) the composition class for the composed residue.

(4) The composition for the composed residues is taken from this rule set. In contrast, for the single residues, each building block is associated with its composition via a dedicated list of rules. In both cases, compositions are assigned when the residue is created. The processed structures populating the resulting knowledge graph are then imported into the triple store.

The data mapping process was implemented through a Python 3.6-based application relying on *Dask* [24] libraries to parallelize structure processing and *RDFLib* [25] to create the RDF resources. The application is designed to be run and to access the data sources remotely using the given configurations. Specific configuration settings are stored in a file with the user’s variables including a URL to locate the GlycoCT source, a base URI to identify uniquely all generated triplets, the number of CPUs and some more processing parameters. Once the GlycoCT is processed and converted to triples, the ontology model, the ontology dependencies, and the RDF statements generated from the data mapping are then loaded into an dedicated instance of a *GraphDB* triple store where the graph is completed by the inference process. The knowledge is then loaded on a test instance and validated using the queries listed in Section 3.3. Once validated, the full graph is transferred to the triple store production instance.

The complete list of the content validation queries that includes the SPARQL code together with the instructions to access the SPARQL endpoint are published on the GlySTreeM wiki page: https://GlyConnect.expasy.org/glystreem/wiki (accessed on 1 November 2021).

## 3. Discussion

Developing the GlySTreeM knowledge base raised questions regarding the soundness of the system as well as the correctness of the knowledge. To address these issues and confidently recommend GlySTreeM to support glycan research, an extensive validation was carried out. The results are presented and discussed below.

### 3.1. Validation Methodology

Validating the different parts of the GlySTreeM knowledge base requires a variegated approach. The semantic representation of the domain should be tested for soundness and expressiveness; these aspects are covered in the model-validation section. The individuals populating the base generated from GlyConnect structures were also examined and the results are discussed in the content-validation section.

### 3.2. Model Validation

The semantic model of the GlySTreeM knowledge base requires validation along two key axes: internal and external consistency. The external consistency requires a reference ontology. The internal consistency validation is focused on the soundness of the axioms on which the model lies and involves checking logical inconsistencies. This task was completed deploying a reasoner on the ontology (without individuals) and analyzing the inferred result using Protegé [26] and the Fact++ [27] reasoner. An inconsistency would cause an entity to be a subclass of the empty set (*owl:Nothing*). A potential issue might be caused by the logical profile of the MODL [23] ontology library included in the model. This profile does not fall in the OWL DL [28] on which both Protege and the reasoner are based. This issue was caused by *modl:treehasAncestor*, *modl:treehasDescendant*, and *modl:treehasSibling*. It was addressed [29] by the removal of irreflexivity axioms of the object properties. The result of the inference on the classes and on the object properties of the ontology does not indicate any inconsistency, thus guaranteeing the consistency of the model.

The consistency of the ontology can be established through its alignment with a higher-level one, in order to verify that the domain modelling does not create semantic inconsistencies (i.e., absurd yet consistent assertions) with respect to a wider domain. To this end, the SKOO ontology [30] representing scientific knowledge was chosen. SKOO was already validated for both internal and external consistency [31] as a result of an alignment with other higher-level, well-established ontologies such as *DOLCE* [32] and *SIO* [33].

The alignment between GlySTreeM and SKOO was performed by creating axioms, listed in Table 2, of type *rdfs:subClassOf*. The ontology with the alignment axioms was then checked for consistency through reasoning, producing no change in the resulting model.

### 3.3. Content Validation

The individuals of the knowledge base represent the glycan structures and their elements. Their validation should target both the correctness of structure expression and the overall quantitative dimension of the base content. Since the individuals were generated from GlyConnect, the validation of the individuals is carried out with GlyConnect as a reference.

#### 3.3.1. Quantitative Queries

The quantitative dimensions of individuals present in the knowledge base was assessed by running a series of SPARQL queries whose results were then compared to those of analogous SQL queries performed on GlyConnect. The following list of queries were run on GlySTreeM and GlyConnect:Count of all structures that only have a substituent in the root node.Count of all glycans that have a ResidueRoot.Count of all unique base types found in ResidueRoot.Count of all unique base types.Count of all unique substituent type.sCount of all unique substituent types that are a single residue (not part of a composed residue).Count of all di-sialyl Tn antigen type structures.Count of all structures that start with Fucose.Count of all structures that start with Mannose.Count of all structures that start with Xylose.Count of all structures that have from (at least) 1 to 9 undefined sections (bag items).Count of all structures that have exactly 1 to 9 undefined sections (bag items).Count of all structures that have no associated GlycoCT string (therefore no GlycanCore).Count of all structures that have associated Glycan.Count of all structures that have associated GlycanCore.Count of all structures that have associated ResidueRoot.

The code for these queries is available on https://glyconnect.expasy.org/glystreem/wiki (accessed on 1 November 2021), and the corresponding results are reported in Table 3. Overall, the matching of GlyConnect data is met with two exceptions explained in the notes below the table.

#### 3.3.2. Qualitative Queries

The correctness of the structure representation is tested through a variety of queries that highlight peculiar structures with complex connections. These queries also have the purpose to test the GlySTreeM model expressivity power and search flexibility (Table 4).

Display IDs for structures matching exactly N-linked GlcNAc2Man3 core.Display IDs for structures with exactly GlcNAc and phosphate.Show the undefined sections (bag items) for structure 671.Does the database contain any structures with two (or more) sialyl acids consecutively linked?Does the database contain any O-linked Core 1 with two sialyl acids consecutively linked?Pull out only O-linked monosaccharides.

An example of a relatively complex query is presented in queries 4 and 5 in Table 4. When presented with a search term that refers to different parts of the structure, in this case, a “motif” (two sialic acid residues linked in-line) and a core type (in this case, an O-linked Core 1), GlySTreeM provides the means to incorporate both aspects in the search (Figure 4).

### 3.4. Benchmark

A number of tools are available for interrogating the structure of glycans; however, most are not directly comparable as they are employed in different contexts, e.g., data mining [34] or mass spectrometric fragment analysis [35]. However, a recently published tool Glycowork 0.2.0 [18] established itself as a means to identify motifs in glycan structures. Hence, there is overlap with the presented method, and this feature can be used in a comparison study. Therefore, in this section, GlySTreeM is bench-marked against the motif-finder part of GlycoWork to assess the functionality in the scope of glycoinformatics and substructure searching.

Glycowork is a standalone Python package and takes as input IUPAC strings of a glycan structure, while our source, the GlyConnect database, contains 4779 structures, all with GlycoCT format but not necessarily with IUPAC. In order to compare the two tools with homogeneous inputs an effort was made to convert GlycoCT to IUPAC using an external tool (GlycanFormatConverter). The IUPAC was validated by normalizing some of the residue names, e.g., NeuAc to Neu5Ac as well as refactoring the linkages with unknown components, e.g., Gal(b1-?) to Gal(bond). The process involved several steps including validation of the GlycoCT, converting to an intermediate format and validating the resulting IUPAC. For this reason, only a subset of the available structures was used as input for the bench-marking.

The Glycowork function motif.annotate.annotate_dataset was used to identify motifs in structures using an inbuilt motif library. The function can take a number of optional parameters, including wildcards. These were set as follows: extra=’wildcards’ and wildcard_list=[’bond’]. The inbuilt motif library within Glycowork includes a leading galactose residue in all the Lewis type motifs. As a final step, Lewis A-, B-, X-, and Y-type motifs without this leading galactose were added to the inbuilt library.

#### 3.4.1. Use Cases

The two tools, GlySTreeM and Glycowork, were used to carry out two tasks that require the identification of two substructures.

##### First Use Case

The first use case required the identification of a Lewis-type substructure in a sequence of residues where the glycosidic linkages may not be fully defined, as in Figure 5a. This was carried out in GlySTreeM by constructing a SPARQL query to search for a defined pattern of residues (Gal[Fuc]GlcNAc[Fuc]).

By omitting the explicit linkages in the sequence all Lewis B and Y containing structures were extracted from GlySTreeM. The set of extracted structures also contain Lewis A and X substructures. The details of the use case execution are summarized in Table 5.

To provide additional context to this use case outcome, the process was extended to five randomly generated structure datasets. The steps taken in this analysis were the following:Generate five random datasets of IDs from GlyConnect.Convert to input formats for each tool.Process input formats with each tool.Produce .svg cartoons for each outcome set.Visual validation of results for positve and negative matches.

The outcome of the analysis is summarized in Table 6.

##### Second Use Case

As shown in the previous use case, structures with undefined sections are extremely relevant not only numerically but also semantically, as they hold most of the ambiguity that can be dealt with by GlySTreeM.

The second use case involved the search for a substructure specifically found in the undefined section of the structure, which poses a more challenging task to resolve. The tools were required to determine if structures with undefined sections could be searched for the presence of a potential Lewis type composition, as in Figure 5b.

GlySTreeM was queried with a SPARQL query that specifically searched for a GlycanBag that contained at least one independent Fuc residue and at least one GlcNAc residue (Figure 5b). The results were manually validated by, again, visually inspecting the .svg representations of the results. Once more, the subset of results from the GlySTreeM output was used as input for the Glycowork package. The details of the use case execution are summarized in Table 7.

## 4. Results

The reported use cases involve searching for Lewis B- or Y-type substructures in the GlyConnect database. Lewis B antigen has been shown to be important in *H. pylori* binding [36], while Lewis Y antigens are highly expressed in epithelial cancers, such as gastric cancer, [37]. On this basis, they constitute a significant search option for potential users of this tool.

The structures annotated from publications, such as those stored in GlyConnect, are solved using many different experimental techniques, some more precise than others. GlySTreeM allows the search of structures with both precisely defined substructures and fuzzy search. The first use case involves the search for Lewis B/Y type structures within a glycan structure. The SPARQL query allowed the specification of the sequence needed, and GlySTreeM gave 116 out of 4479 positive matches for Lewis B/Y substructure. The output from GlySTreeM was manually validated as containing Lewis-type structures by visualizing the .svg representations. This set of 116 structures was then used as a test set for GlycoWork.

Not taking into account the GlycoCT to IUPAC conversion, which is a pre-requisite for this comparison, Glycowork needed several iterations to look for fuzziness. This involved specifically tagging the linkages as unknown by using the wildcard “bond” in the search. It was also realised that GlycoWork motif library contains Lewis-type structures with a leading galactose residue. This definition of the motif resulted in a number of false negatives. By customising the motif library, the number of false negatives was reduced to zero, with just three cases of invalid IUPAC codes not being processed (Figure 6a). The output was manually validated by searching for matches to Lewis A,B,X or Y as defined in the motif library.

To more deeply analyse this use case outcome, additional statistical context is provided. In particular, the search for Lewis B-/Y-type structures was performed on five randomly generated datasets to account for potential false positives and negatives.

The outcomes from the two methods are very similar. GlycoWork gives only one false negative in dataset 03 (GlyConnect structure ID 3529).

Almost 30% of the dataset was not processed in GlycoWork due to the presence of UND section (GlyConnect ID 202) or residues not handled such as internal phosphate bonds (GlyConnect ID 1739) or Delta4 (GlyConnect ID 3005). This seems to confirm identified limitations of the IUPAC condensed format in dealing with structural ambiguity or non/rare monosaccharides [10,38]. Nonetheless, this is significant given that more than one quarter of the GlyConnect database consists of structures that contain undefined (UND) sections.

The second use case involved the search for a substructure specifically found in the undefined/GlycanBag section of the structure, which poses a more challenging task to resolve.

GlySTreeM proved its effectiveness in this case by allowing the search using incomplete information in a undefined structure domain such as the one represented in Figure 6b. The task was accomplished without additional effort with respect to the search with defined substructure. The output from GlySTreeM was manually validated as containing Lewis-type structures by visualizing the .svg representations.

In an effort to exploit the available features, several iterations of GlycoWork were performed, but the incomplete structures were not processed by the package. Logically, no output was available to validate.

This reflects the different purposes of the two methods. The common goal of identifying substructures/motifs is efficiently reached by both. The requirement for dealing with ambiguous structures, initially stated as our goal, is met by GlySTreeM.

## 5. Conclusions

Experimental techniques, in particular mass spectrometry, are now commonly used to solve glycan structures, either attached to (glycoproteomics) or released from (glycomics) glycoconjugates. As a result, the degree of precision of glycan structures has spread, over time, as data are deposited in GlyTouCan. Thousands of entries are ambiguously defined with a variety of cases where one or the other or several linkages are unknown. A solution for navigating in this large set of related data was proposed with the GNOme ontology, but it does not allow substructure search and glycan comparison. A few years ago, we suggested an RDF-based model for mapping substructures in full structures, and we built on this experience to define a new ontology with a tree logical structure that captures more faithfully the specificity of glycan structural data. This resulted in GlySTreeM, the first version of which was introduced here. GlySTreeM was validated with GlyConnect data but it can process any data source provided glycan structures are encoded in GlycoCT; when other formats are used, a translation is easily obtained with GlycanFormatConverter. GlySTreeM does not rely on the predefinition of glycan motifs, and any structural fragment can be searched in the selected glycan source. Furthermore, the tool handles glycan compositions (no linkage information at all). At present, incoming data mostly arise from higher eukaryote species thus limiting the residue set to a few dozens. We envisage the expansion of the model to accommodate procaryote glycans that span a much wider and poorly chartered residue space. The rule system can also be amended if or when needed. GlySTreeM will be fully integrated in GlyConnect to improve the consistency of the database. It will also be used to cross-reference glycan ligands of SugarBindDB [39] and UniLectin [40] with full structures of GlyConnect. Finally, we plan to offer a standalone service for substructure search in replacement of the previously available but outdated GlyS3 (Available online: https://glycoproteome.expasy.org/substructuresearch (accessed on 1 November 2021)).

## Figures and Tables

**Figure 1 molecules-27-00065-f001:**
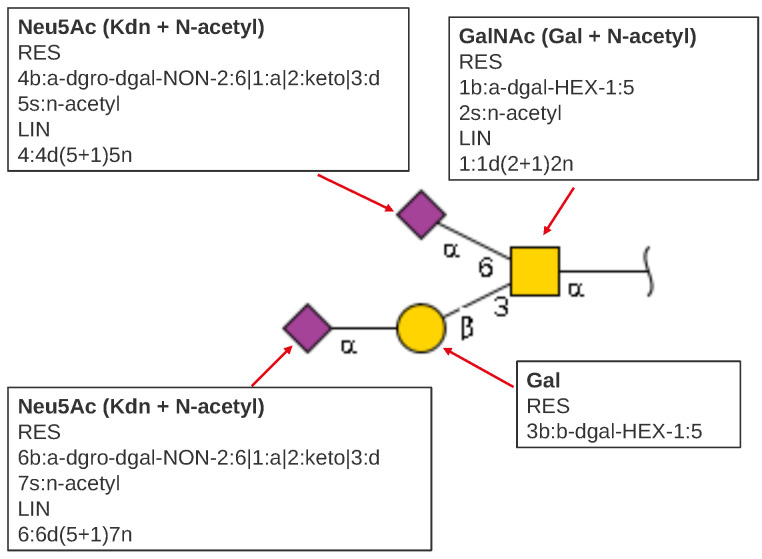
GlycoCT Breakdown.

**Figure 2 molecules-27-00065-f002:**
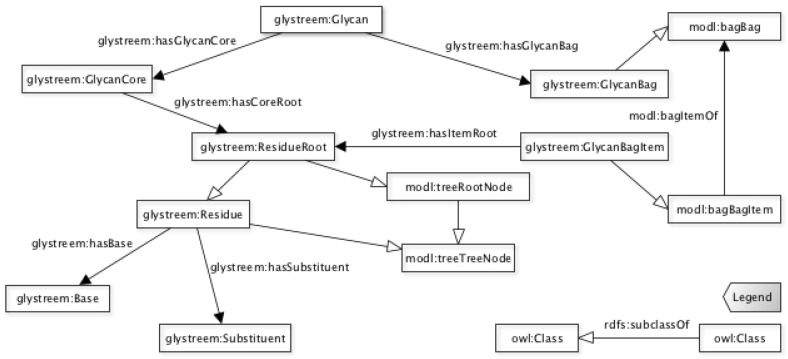
The class hierarchy describing the glycan model top classes and the residue structure of the core and the bag.

**Figure 3 molecules-27-00065-f003:**
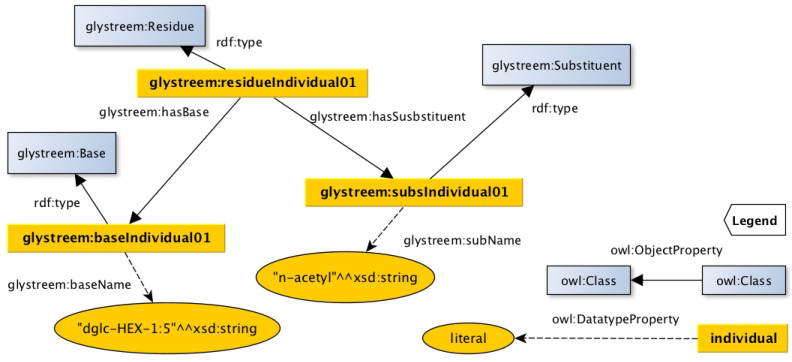
Semantic representation of the GlcNAc building block in the GlySTreeM knowledge base: the base and the substituent are related to the same residue.

**Figure 4 molecules-27-00065-f004:**
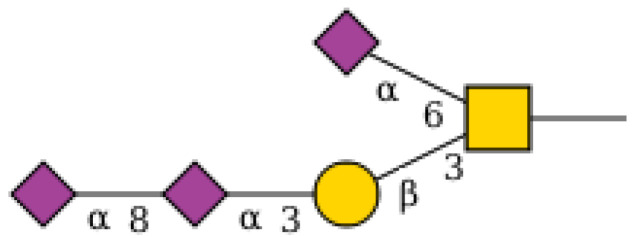
Example of a structure extracted from GlySTreeM using SPARQL query.

**Figure 5 molecules-27-00065-f005:**
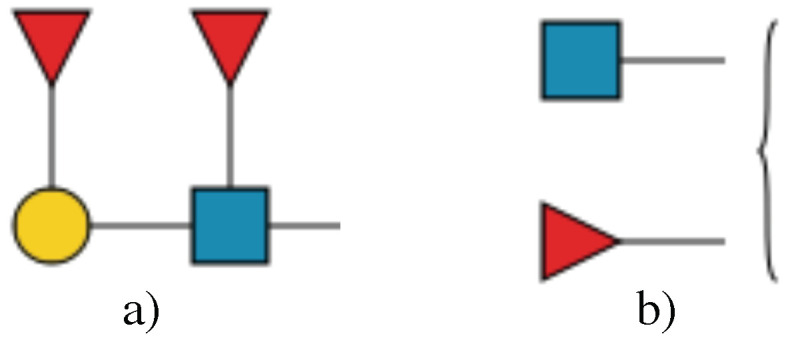
(**a**) Lewis B/Y type structure and (**b**) Free Fuc and GlcNAc residues

**Figure 6 molecules-27-00065-f006:**
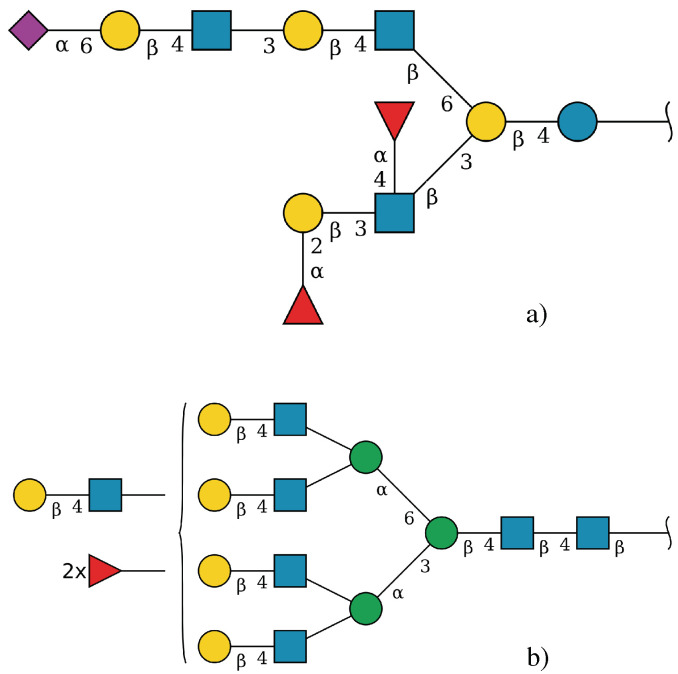
Example of false-negative structure assigned by Glycowork. (**a**) Example of false negative structure assigned by Glycowork due to an IUPAC validation error and (**b**) Example of an undetermined structure potentially containing the Lewis B or Y motifs.

**Table 1 molecules-27-00065-t001:** Textual and graphical representation of GlcNAc using IUPAC and GlycoCT syntaxes.

IUPAC		GlycoCT		SNFG Cartoon
GlcNAc		RES 1b:x-dglc-HEX-1:5 2s:n-acetyl LIN 1:1d(2+1)2n		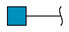

**Table 2 molecules-27-00065-t002:** Correspondences between GlySTreeM and SKOO classes.

GlySTreeM Class	SKOO Class
Glycan	⊑ DomainObject
Residue	⊑ DomainObject
Molecule	⊑ DomainObject
Base	⊑ DomainObject
Substituent	⊑ DomainObject
Epitope	⊑ Hypothesis
GlycanCore	⊑ Proof
GlycanBag	⊑ Hypothesis
GlycanBagItem	⊑ Observation
ResidueRoot (in core structures)	⊑ Assertion

**Table 3 molecules-27-00065-t003:** Quantitative Queries—Results.

Query No.	GlyConnect Result	Triple Store Result	Notes
1	0	0	Expected result
2	4781	4779	Note 1
3	20	20	Expected result
4	39	39	Expected result
5	6	6	Expected result
6	5	5	Expected result
7	9	9	Expected result
8	10	10	Expected result
9	82	82	Expected result
10	4	4	Expected result
11	1285, 809, 477, 242, 111, 56, 23, 8, 2	1285, 809, 477, 242, 111, 56, 23, 8, 2	Expected result
12	476, 332, 235, 131, 55, 33, 15, 6, 2	476, 332, 235, 131, 55, 33, 15, 6, 2	Expected result
13	29	29	Expected result
14	4810	4808	Note 1 and 2
15	4781	4779	Note 1 and 2
16	4781	4779	Note 1 and 2

Note 1 GlyConnect contains two structures with repeat units (structures 414 and 2371). These were omitted from
the triple store. Note 2 The total number of structures includes those with GlycoCT strings (4779) and those
without (29). Only those with GlycoCT strings have a GlycanCore or a ResidueRoot. Average real time for these
queries varied between 0.07 and 0.60 s.

**Table 4 molecules-27-00065-t004:** Qualitative Queries—Results.

Query No.	GlyConnect Result	Triple Store Result	Notes
1	Yes—11, 9901	Yes—11, 9901	Expected result
2	3316	3316	Expected result
3	View cartoon (id 671 3 undefined Gal residues)	3 sections; bagitem 1, bagitem 2, bagitem 3	Expected result
4	51 structures	51 structures	Expected result
5	Yes, structure 3456	Yes, structure 3456	Expected result
6	O-linked monosaccharide subset View dataset	Same subset of structures	Expected result

Average real time for these queries varied between 0.18 and 0.68 s.

**Table 5 molecules-27-00065-t005:** Use Case 1.

**Description**	Identify Lewis A/B/X/Y type substructures.
**Actor**	Bioinformatician/Scientist.
**Initial Conditions**	List of 4779 glycans^1^ in GlycoCT.
**Actor Actions**	**Systems Response**
(1) SPARQL query on GlySTreeM searching for Lewis A/B/X/Y substructure.	(2) 116 positive matches.
(3) Produce 116 IUPAC codes ${}^{2}$.	(4) 114 IUPAC ${}^{3}$.
(5) Run annotate-glycan function on 114.	(6) Glycowork: 53 Positive, 28 negative, 33 IUPAC validation error.
(7) Validate IUPAC and rerun.	(8) Glycowork: 53 Postive, 61 negative.
(9) Added wildcards to optional arguments.	(10) Glycowork: 64 positive, 50 negative.
(11) Added customised motifs.	(12) Glycowork: 102 positive, 12 negative.
(13) Refactored IUPAC condensed for 12 false negatives	(14) Glycowork: 112 positive, 1 false negative (ID 3529) and 1 IUPAC validation error (see Figure 6a).
**Post Conditions**	
The 116 positive results from GlySTreeM were manually validated by inspecting corresponding .svg representations. After several iterations of Glycowork on 116 subset of structures for searching of Lewis X/Y/A/B substructures and gradual fine-tuning, there remained three that could not be validated due to IUPAC formatting errors (see Figure 6a).	

^1^ Data from GlyConnect; ^2^ 71 available in GlyConnect, 33 were converted using GlycanFormtConverter; ^3^ 2 were
discounted as they gave GlycoCT validation error with the converter.

**Table 6 molecules-27-00065-t006:** Use Case 1—Randomised dataset analysis.

	Total	Converted	Processed	Positives	TP	FP	TN	FN ${}^{1}$
**DS-01**								
**GlySTreeM**	200	199	199	8	8	0	191	0
**Glycowork**	200	152	146	8	8	0	138	0
**DS-02**								
**GlySTreeM**	200	199	199	4	4	0	195	0
**Glycowork**	200	152	146	4	4	0	142	0
**DS-03**								
**GlySTreeM**	200	198	198	3	3	0	195	0
**Glycowork**	200	141	135	2 ${}^{2}$	1	0	133	1 ${}^{3}$
**DS-04**								
**GlySTreeM**	200	199	199	6	6	0	193	0
**Glycowork**	200	136	131	6	6	0	125	0
**DS-05**								
**GlySTreeM**	200	198	198	6	6	0	192	0
**Glycowork**	200	135	130	6	6	0	124	0

^1^ Abbrev: DS, DataSet; TP, True Positives; FP, False Positives; TN, True Negatives; FN, False Negatives. ^2^ One
of the true positives was not converted to IUPAC with GlycanFormatConverter so is not present in the dataset
processed by GlycoWord. ^3^ GlycoWork false negative is structure ID 3529 in GlyConnect.

**Table 7 molecules-27-00065-t007:** Use Case 2.

**Description**	Identify Lewis A/B/X/Y type substructures in GlycanBags (undefined).
**Actor**	Bioinformatician/Scientist.
**Initial Conditions**	List of 4779 glycans ${}^{1}$ in GlycoCT.
**Actor Actions**	**Systems Response**
(1) SPARQL query on GlySTreeM searching for Lewis A/B/X/Y.	(2) 11 positive matches.
(3) Produce 11 IUPAC codes.	(4) 7 IUPAC ${}^{2}$.
(5) Run annotate-glycan function on 7.	(6) Glycowork: IUPAC validation error.
(7) Validate IUPAC and rerun.	(8) Glycowork: IUPAC validation error.
(9) Added wildcards to optional arguments.	(10) Glycowork: IUPAC validation error.
(11) Added customised motifs.	(12) Glycowork: IUPAC validation error (see Figure 6b).
**Post Conditions**	
The 11 structures that were identified by GlySTreeM all contain at least one Fuc residue and one GlcNAc in the GlycanBag, as validated by manually inspecting the SVG representations. Using the features introduced in the previous use case, no result other than IUPAC validation error was given by Glycowork.	

^1^ Data from GlyConnect; ^2^ 3 had no available IUPAC in GlyConnect; 1 did not have a Gal attached to the GlcNAc.

## Data Availability

https://glyconnect.expasy.org/glystreem/wiki (accessed on 1 November 2021).

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
