# Peer review of "Dealing with the Ambiguity of Glycan Substructure Search"

_molecules, 2021, doi:10.3390/molecules27010065_

Round 1
Reviewer 1 Report
Review report
With the recent progress of glycomics, many glycan structures with ambiguous and incomplete information on linkage positions, anomers, etc. have been registered in repositories. The authors have found a data structure that allows for more appropriate encoding and easier retrieval of substructures and motifs than the conventional ones. Using the recently proposed ontology for tree structures, they defined an ontology for glycan structures and converted those described by GlycoCT into RDF. The resultant RDF triples were registered as SPARQL endpoints and made searchable. The internal verification of the data structure was done using the data of GlyConnect, whose nature is well known by authors. We also demonstrated its usefulness by comparing it with the search in GlycoWorks. This paper is worthy of publication because of its value in enabling motif retrieval through a data structure using a tree ontology. This reviewer hopes that the following comments will be considered and the paper will be of higher quality.
General concept comments
In the SPARQL search experiment, the reviewer thinks that the test data is too small because the search target is at most 5000 records. Of course, the reviewer believes that it is natural and useful to perform searches on small, well-understood data sets for data structure verification. However, SPARQL searches are generally slow to execute, making searches on large data sets impractical. For example, if the authors construct a dataset by converting the possible data of GlyTouCan to GlySTreeM, they can obtain about 100,000 data. How long would it take to run the same search experiment as the one in the manuscript? Wet users want to know what functions have been reported for the motifs obtained by their experiments, what biological origin they have, and so on. Therefore, the database to be searched should be large. The reader would like to know how effective SPARQL search is for such a large database. This reviewer would like to see appropriate additional experiments for large datasets in this regard.
GlySTreeM is an innovative encoding scheme, but it requires data import from GlycoCT. In addition, there are many encoding schemes in the field of glycan informatics. The conversion between them must be done by the user. Furthermore, if the user does not understand SPARQL and cannot write queries, the search cannot be performed. This may be obvious to informatics developers, but this reviewer thinks it is a huge obstacle for the average wet researcher. The reviewer hopes that the authors will show in the manuscript what they think about this point and encourage general wet researchers to utilize this method. When replacing GlySTreeM with GlyS3, the development of this software will be more valuable if it supports various coding schemes and the conversion of coding schemes is handled internally.
This reviewer feels that the use of words such as “structure” and “molecule” is not common. For example, is it correct that structure and molecule in line 103 are glycan and residue, respectively? Also, “N-acetyl” in line 109 is a group, not a molecule. If the authors use the term “structure” and “molecule”, they should include the definitions in the text.
Specific comments
The introduction consists of one paragraph, which is long, and should be divided into three or four paragraphs. The contents seem to change at lines 21 and 57, so at least they should be split into paragraphs.
In line23-25, if the glycans referred to by the authors are N-linked glycans, O-linked glycans, or GSL glycans, it is wrong that glucose is frequently included in the cyclic building blocks. GlcNAc would be more appropriate.
In section 3.3.1 and 3.3.2, the reviewer thinks that it would help the reader to understand better if one or two specific SPARQL sources were included in the text for the queries of search tests to show how to search for the substructure. The reviewer thinks that the query 4 or 5 in section 3.3.2 is appropriate. Showing exactly how the Lewis structure is represented in SPARQL will help readers understand it better.
Reviewer 2 Report
In the present manuscript, Daponte V., et al. introduce a new ontology with tree logical structure to represent glycan structures, that would also allow for querying structural fuzziness. This work thus addresses a long-standing issue in the fields of glycobiology and its need for tools towards meaningful glycan database usage. For validation of their concepts, the authors implemented the GlySTreeM knowledge base by importing all GlycoCT-encoded data from the GlycConnect database, and present results of various use-cases. The manuscript is very well written and scientifically sound. Prior to publication, I recommend the authors to address the following points:
major point:
The authors present results of two awkward bench marking tests/use-cases against Glycowork. While I do appreciate the authors efforts to produce a comparison to existing tools, it remains unclear if -or in how far- the dreadful performance of Glycowork is merely a result of the conversion of the GlycConnect data from GlycoCT to IUPAC-condensed data format. Please elaborate and discuss possible reasons for this poor performance of Glycowork.
The use cases presented for the qualitative validation process are ill-defined and the authors evaluation of the results is sometimes unclear. E.g.
qual query 3: What exactly are the undefined sections in structures 671? The galactose residues?
qual query 5: How exactly was the O-linked Core 1 structure defined in the search?
qual query 6: How is the O-glycosylation defined/inferred/encoded in GlyConnect/GlycoCT? How many structures (number) were retrieved?
I suggest the authors to more clearly define their search parameters or filtering criteria, and to discuss their results more thoroughly.
minor points:
Comparison of the data-base sizes of GlyConnect and Triple Store/GlyStreeM knowledge base?
line 163: residue representation of the GlcNac molecule (change to GlcNAc)
line 237: The consistency, of the ontology (remove ",")
line 284: Display IDs for structures with exactly GlcNAC (change to GlcNAc)
Round 2
Reviewer 2 Report
The authors have responded to most of my points and have adapted the manuscript accordingly. (The present version of the manuscript lacks all reference and figure numbers; presumably a formatting error.)
The use case of searching for Lewis substructures still is strange: First, the authors use their approach to identify a subset of Lewis-type glycan structures from GlyConnect, which they then use as input for a different software to compare GlySTreeM with. The example presented in the manuscript does not provide any information on: a) how many Lewis-type glycan structures in the data-base were not found by GlySTreeM, and b) how many of these "missing" Lewis-containing structures would have been detected by Glycowork. I think this would be important. The second use-case does not result in any meaningful output by Glycowork; presumably due to format conversion issues. As a consequence, from the use-case results presented in the manuscript, the reader is merely left with the impression that Glycowork searches are complicated to execute and -in most cases- useless. Would a different use case (e.g. find core-fucosylated sturctures, or -intriniscally terminal- sialyl Lewis X structures) yield more comparable results by the two approaches?
